# Molecular Approaches Using Body Fluid for the Early Detection of Pancreatic Cancer

**DOI:** 10.3390/diagnostics11020375

**Published:** 2021-02-22

**Authors:** Kennichi Satoh

**Affiliations:** Division of Gastroenterology, Tohoku Medical and Pharmaceutical University, 1-15-1 Fukumuro, Miyaginoku, Sendai, Miyagi 983-8536, Japan; ksatoh@tohoku-mpu.ac.jp; Tel.: +81-22-259-1221; Fax: +81-22-290-8959

**Keywords:** pancreatic ductal adenocarcinoma, early detection, gene mutation, liquid biopsy

## Abstract

Pancreatic ductal adenocarcinoma (PDAC) is the most malignant form of gastrointestinal tumor and is the fourth leading cause of deaths due to cancer in Japan. This cancer shows a poor outcome due to the difficulty of its early diagnosis and its rapid growth. Once this disease becomes clinically evident, it is frequently accompanied by distant metastasis at the time of diagnosis. A recent multicenter study in Japan revealed that patients with the early stage of this disease (stage 0 and I) showed favorable prognosis after surgical resection, indicating the importance of early detection for improvement of PDAC prognosis. PDAC develops through a stepwise progression from the precursor lesion, and over the last few decades molecular analyses have shown the detailed genetic alterations that occur in this process. Since advances in molecular technologies have enabled the detection of genetic changes from a very small quantity of samples, a large number of non-invasive molecular approaches have been utilized in an attempt to find precursor or non-invasive carcinoma lesions. In this review, the current efforts in terms of the molecular approaches applied for the early detection of PDAC—especially using body fluids such as pancreatic juice, blood, and saliva—are summarized.

## 1. Introduction

Pancreatic ductal adenocarcinoma (PDAC) is the fourth leading cause of deaths due to cancer in Japan [1]. This disease usually shows a poor prognosis because of its rapid progression and the development of distant metastasis at the time of diagnosis. In addition, this tumor is resistant to conventional chemotherapy and radiation therapy. Approximately 40,000 people were diagnosed with PDAC and 35,000 people died from this disease in 2018 in Japan [1]. The 5-year relative survival rate, for all patients with PDAC combined, was found to be less than 10% [1]. On the other hand, a multicenter study demonstrated that the estimated overall survival rates at 10 years after resection for stage 0, stage I (Ts1a) and stage I (Ts1b) PDAC were 94.7, 93.8 and 78.9%, respectively, although the prevalence of stage 0 and I PDAC patients was as low as approximately 3% [2]. Therefore, new techniques for the early diagnosis, and/or a therapeutic strategy for the advanced stage of the disease, are required in order to improve the outcome of PDAC. 

In the last few decades, molecular analyses have revealed detailed information about the gene alterations involved in tumorigenesis and/or the development of PDAC [3,4,5,6]. Since recent advances in molecular technologies have enabled the accurate detection of genetic changes from a small quantity of samples, various molecular approaches have been used in an attempt to diagnose PDAC using small amounts of cells, obtained by endoscopic ultrasound fine-needle aspiration (EUS-FNA) or brushing cytology during endoscopic retrograde cholangiopancreatography (ERCP). However, these analyses have been limited due to the procedural risks of collecting samples and intra-tumor heterogeneity. The goal of molecular approaches is to detect the premalignant lesions that develop into invasive cancer using non-invasively obtained materials such as urine, oral fluid (saliva), peripheral blood or pancreatic juice, when imaging examination is not feasible. This review focuses on recent molecular approaches used for the early detection of PDAC via liquid biopsy (using body fluids including pancreatic juice, peripheral blood and saliva), and summarizes key research.

## 2. Gene Mutation Analysis in Pancreatic Juice

PDAC develops through a stepwise progression from pancreatic intraepithelial neoplasia (PanIN), with an accumulation of genetic mutations [3,4,5,6]. A large number of genetic alterations have been shown to be involved in this process, with the most common genetic alterations of pancreatic carcinogenesis including oncogene (*KRAS*) and tumor suppressor gene mutations (*CDKN2A, TP53* and *SMAD4*) [5,6]. Approximately 90% of PDAC cases harbor *KRAS* mutations and alterations of *CDKN2A*, almost 70% contain *TP53* mutations and approximately 20% harbor *SMAD4* mutations [5,6]. *KRAS* mutations emerge in PanIN-1, *CDKN2A* mutations occur in PanIN-1/2, *TP53* alterations generally occur with high grade dysplasia (i.e., PanIN-3) and the inactivation of *SMAD4* emerges in lesions with high-grade dysplasia, promoting a transition to invasive cancer [4,5,6]. The collection of pancreatic juice from the duodenal lumen is less invasive than ERCP and/or EUS-based cell collection methods, and cells can be obtained directly from the pancreatic duct. Thus, a large number of molecular assessments for the diagnosis of PDAC using pancreatic juice have been performed. Since *KRAS* gene mutations frequently occur from low grade PanIN [4,5], such mutations have been analyzed in pancreatic juice for the early detection and/or diagnosis of PDAC using polymerase chain reaction (PCR)-based methods for about 30 years. However, the analysis of mutations in this gene has shown a low level of specificity for the diagnosis of PDAC because such mutations are also detected in non-malignant pancreatic tissues, such as chronic pancreatitis (CP) [7,8,9,10]. A meta-analysis of 16 studies assessing the diagnostic value of the detection of *KRAS* mutations revealed that the sensitivity and specificity levels for the diagnosis of PDAC were 59 and 87%, respectively [11]. To improve the diagnostic accuracy, additional gene analyses, for example, investigating *TP53*—alone, or in combination with *KRAS* mutation analysis and conventional cytological examination or analysis of telomerase activity—have been studied [12,13,14,15,16,17]. A meta-analysis of 39 studies assessing the diagnostic utility of the four major altered genes in PDAC (*KRAS, CDKN2A, TP53* and *SMAD4*), telomerase activity, and combination assay using pancreatic juice, demonstrated that the most reliable marker was telomerase activity. The sensitivity and specificity of telomerase activity were 82 and 96%, respectively, while the sensitivity and specificity of *KRAS* were 67 and 82%, respectively. On the other hand, three tumor suppressors (*CDKN2A*, *TP53* and *SMAD4*) showed low sensitivity (35, 53 and 36%, respectively) [18]. However, these molecular approaches are limited in that they cannot distinguish early PDAC from IPMN, or clinically normal pancreas functioning with low-grade PanIN, since alterations of *KRAS* and other tumor suppressors, in addition to telomerase activity are also found in these lesions. 

Recently, next-generation sequencing (NGS) technology has allowed researchers to sequence DNA much more quickly at relatively low costs [19]. Yu et al. employed a novel NGS method, termed digital NGS, to detect low concentrations of gene mutations in pancreatic juice from 115 subjects (34 cases with PDAC, 57 cases with IPMN cases and 24 controls) [20]. They explored nine genes (*KRAS, GNAS, TP53, SMAD4, CDKN2A, RNF43, TGFBR2, BRAF* and *PIK3CA*) that are frequently mutated in PDAC and/or IPMN in pancreatic juice. The prevalence of mutations of *KRAS, TP53, SMAD4* and *GNAS* in the pancreatic juice of patients with PDAC was 74, 59, 15 and 38%, respectively. In pancreatic juice from IPMN patients, the prevalence of mutations of *KRAS, TP53, SMAD4* and *GNAS* was 75, 26, 2 and 53%, respectively. In pancreatic juice from controls, mutations of *KRAS* and *GNAS* were found in 42 and 17% of subjects, respectively, while no mutations of *TP53* or *SMAD* were detected. In, addition, mutant DNA concentrations were higher in patients with PDAC than IPMN or controls. Interestingly, mutant *TP53/SMAD4* concentrations could distinguish PDAC from IPMN with 100% specificity. Two cases had *SMAD4* or *TP53* mutations detectable in their pancreatic juice samples over 12 months prior to their pancreatic cancer diagnosis, at a time when no suspicious lesions were detected by imaging, suggesting the possibility of the early detection of PDAC using this technology [20]. Suenaga et al. also demonstrated that mutant *TP53/SMAD4* concentrations in pancreatic juice are useful to distinguish patients with PDAC or high-grade dysplasia from all other subjects using the digital NGS [21]. They examined 12 gene mutations (*KRAS, GNAS, TP53, SMAD4, CDKN2A, RNF43, TGFBR2, BRAF, PIK3CA, ARID1A, FBXW7* and *VHL*) in pancreatic juice from 67 patients undergoing pancreatic evaluation, including patients with PDAC (*n* = 14), patients who subsequently underwent pancreatic resection for precursor lesions (*n* = 13), patients undergoing surveillance for their familial/inherited susceptibility to PDAC (*n* = 31) and normal pancreas disease control (*n* = 9). They revealed that the concentration of total gene mutations in pancreatic juice differentiated patients with PDAC or high-grade dysplasia in their resected specimen from all other subjects with 72.2% sensitivity and 89.4% specificity (area under the receiver operator characteristic curve (AUC) = 0.872), and that the concentrations of mutant *TP53/SMAD4* were able to distinguish patients with PDAC or high-grade dysplasia in their resected specimens from all other subjects with 61.1% sensitivity and 95.7% specificity (AUC = 0.819). Therefore, *TP53/SMAD4* concentrations measured by digital NGS could be an effective tool for identifying pancreatic preinvasive lesion and/or a predictor of PDAC derived from IPMN, since this concentration distinguished PDAC from IPMN.

## 3. Circulating Tumor DNA

### 3.1. Gene Alteration

Circulating cell-free DNA (cfDNA) was first described by Mandel and Metais in 1948 [22]. Then, 30 years after the discovery, the level of cfDNA was shown to be elevated in over half of cancer patients [23]. Molecular analysis of cfDNA revealed that cfDNA from cancer patients contains tumor-associated mutations and can be used for cancer diagnostics and follow-up [24]. In cancer patients, a portion of cfDNA is tumor-derived and is termed circulating tumor DNA (ctDNA). Since the *KRAS* gene is most commonly mutated (approximately 90%) in PDAC tissues [4,5,6], this gene alteration has been the focus in much of the cfDNA analysis for patients with PDAC. Takai et al. investigated *KRAS* mutation status in plasma cfDNA using multiplex droplet digital PCR (ddPCR) in 259 PDAC patients [25]. A multiplex ddPCR assay was created to detect the four common *KRAS* mutations (G12D, G12V, G12R and G13D), which account for 90% of all *KRAS* mutations in PDAC. Among 151 inoperable PDAC patients, mutant *KRAS* was detected in 63 of 107 (58.9%) patients with distant organ metastasis, while only 8 of 44 (18.2%) patients without distant metastasis showed detectable levels of *KRAS* mutation. On the other hand, 9 of 108 (8.3%) patients with resectable PDAC had detectable levels of *KRAS* mutation in plasma cfDNA, indicating that this approach is unlikely to be suitable for the early detection of PDAC [25]. Allenson et al. examined *KRAS* mutation at codons 12/13 in cfDNA from 68 PDAC patients and in 54 healthy controls by ddPCR [26]. They found *KRAS* mutation in cfDNA in 45.5% of patients with localized cancer, 57.9% of those with metastasis and 14.8% of healthy controls [26]. Maire et al. also showed *KRAS* mutation in serum cfDNA from non-cancer controls such as patients with chronic pancreatitis [27]. They examined *KRAS* codon 12 mutations in cfDNA from serum in 47 patients with histologically proven PDAC and 31 controls with chronic pancreatitis using PCR and allele-specific amplification. *KRAS* mutations were found in 22 patients with PDAC (47%) and in 4 controls with chronic pancreatitis (13 %), none of whom developed a PDAC within the 36 months of follow-up [27]. In contrast, Cohen et al. detected *KRAS* mutation in 66 of 221 (30%) of cfDNA in patients with resectable PDAC (Stage I and II) but in only 1 of the 182 plasma samples from the control patients without known cancer using PCR-based assay that could simultaneously assess the two codons (codon 12 and 61), called the “Safe-Sequencing System” [28]. In addition, the authors revealed that the use of *KRAS* in conjunction with four protein biomarkers (CA19-9, CEA, HGF and OPN) increased the sensitivity to 64% for diagnosis of PDAC [28]. This combination assay of gene mutation and protein biomarkers in blood was developed by the same group and applied to 1005 patients with 8 nonmetastatic, clinically detected types of cancers including pancreatic (*n* = 93: 4 at stage I; 83 at stage II; 6 at stage III) [29]. They selected 16 genes (*NRAS, CTNNB1, PIK3CA, FBXW7, APC, EGFR, BRAF, CDKN2A, PTEN, FGFR2, HRAS, KRAS, AKT1, TP53, PPP2R1A* and *GNAS*) and 8 proteins (CA125, CEA, CA19-9, Prolactin, HGF, OPN, Myeloperoxidase and TIMP-1) and examined whether combinations of these in blood tests could diagnose resectable cancers. The presence of a mutation in an assayed gene or an elevation in the level of any of these proteins would classify a patient as positive. This test, called CancerSEEK, showed high sensitivity to the detection of PDAC (approximately more than 70%) and high specificity (greater than 99%: 7 of 812 controls judged as positive). CancerSEEK is not specific to PDAC, but it is likely to be good non-invasive application for screening of high-risk individuals of PDAC, although further prospective studies in a large population will be required. The usefulness of serum protein biomarker has been also shown by Mellby et al. They demonstrated that a serum biomarker signature consisting of 29 proteins could distinguish patients with stage I or II from controls (AUC = 0.96) [30].

On the other hand, Okada et al. demonstrated high diagnostic performance for early detection of PDAC using a PCR-based method. They evaluated KRAS and GNAS mutation in cfDNA from 96 PDAC, 112 IPMN and 76 controls using pre-amplification digital PCR. This method could detect *KRAS* mutation in plasma from early-stage PDAC at a higher frequency than methods used in previous studies: 27 of 38 (71.1%) stage II and III; 2 of 4 (50%) stage 0 patients; and 6 of 9 (66.7%) stage I patients [31]. The authors postulated that the high detection sensitivity for stage 0-I tumor cfDNA was due to relatively rich tumor cellularity, since most of the stage 0-I tumors that they examined were IPMN-associated PDAC.

In order to avoid false-positive results in cfDNA measurements, Macgregor-Das et al. performed digital NGS after enzymatic treatment for plasma DNA to limit chemical modifications of DNA that yield false-positive mutation calls [32]. They measured plasma *KRAS* and *GNAS* hotspot mutation levels in 140 subjects, including 67 patients with PDAC and 73 healthy and disease controls. Positive digital NGS scores indicating the presence of *KRAS* mutations were detected in 23 of 63 (36.5%) patients with PDAC. Among PDAC patients, 14 of 45 (31.1%) stage I/II and 9 of 18 (50%) stage IV cases had a positive score for *KRAS*, while only one case had a positive score for *GNAS*. Further, 2 of 71 (2.8%) disease control patients showed a positive digital NGS score; however, no incidences of PDAC occurrence were observed in these two cases at follow-up evaluation more than 2 years later. Detection of low-level, true positive cfDNA was limited by frequent low-level detection of false-positive calls in plasma DNA from controls. Due to this limitation, this approach could not increase the diagnostic sensitivity of cfDNA detection compared to other previous methods. 

This evidence suggested that the detection of *KRAS* mutation in cfDNA would be limited for early detection of PDAC due to its low sensitivity to the early stage of the disease; however, this approach was shown to be more valuable as a prognostic biomarker [33].

### 3.2. Epigenetic Alteration

DNA methylation is a pivotal mechanism for regulating gene expression in both normal and tumor cells. It occurs by the addition of a methyl group to DNA on a wide range of CpG islands. When a CpG island in the promoter region of the tumor suppressive gene is methylated, expression of the gene is repressed and the function as a tumor suppressor is reduced [34]. Recently, methylation status could be detected in patients’ blood, and this approach have been applied to detect early PDAC. To identify novel DNA methylation biomarkers in PDAC, Yi et al. performed transcriptome-wide microarray screening in four pancreatic cancer cell lines and detected a total of 1427 unique genes as candidate hypermethylated genes in these cell lines [35]. Then, they filtered down to genes that only showed cancer-specific methylation, and selected eight genes (*TFPI2, ASCL2, BNC1, TWIST1, BNIP3, ADAMTS1, PNMT* and *EVL*). Among these, *BNC1* and *ADAMTS1* were verified as potential biomarkers for early detection of PDAC by examination of the methylation status in 123 paraffin-embedded PDAC tissues using methylation-specific PCR (MSP). The two genes demonstrated high methylation frequency from PanIN to advanced PDAC tissues. Using nanoparticle-enabled methylation on beads (MOB) techniques, these alterations were investigated in serum samples from 42 patients with PDAC and 26 controls. The results showed sensitivity to *BNC1* of 79% and to *ADAMTS1* of 48%, while specificity was 89% for *BNC1* and 92% for *ADAMTS1*. Interestingly, sensitivity of detection of stage I PDAC was 90% for both genes. The applicability of this two-gene panel as a biomarker set for early detection of PDAC was confirmed by the same group in independent cohorts [36]. Blood samples from PDAC (*n* =39), pancreatitis (*n* = 8) and control groups (*n* = 95) were evaluated. Methylation of *ADAMTS1* was detected in 87.5% of stage I, 77.8% (7/9) of stage IIA and 90% (18/20) of stage IIB patients with PDAC. *BNC1* methylation was found in 62.5% (5/8) of stage I, 55.6% (5/9) of stage IIA and 65% (13/20) of stage IIB patients. The combination of the two genes showed positive methylation in blood from 100% (8/8) of stage I, 88.9% (8/9) of stage IIA and 100% (20/20) of stage IIB patients with PDAC, with a combined sensitivity of 97.3% and a combined specificity of 91.6%. On the other hand, Shen et al. examined large-scale epigenetic alterations in plasma from various carcinoma patients including PDAC [37]. They developed a sensitive, immunoprecipitation-based method called “cell-free methylated DNA immune-precipitation and high-throughput sequencing (cfMeDIP-seq)” to investigate the genome-wide plasma DNA methylation profiling. This method can increase CpG-rich, potentially more informative fragments and showed a large number of alterations in DNA methylation that are enriched for tumor-specific patterns. First, they generated cfMeDIP-seq profiles from plasma cfDNA of 24 patients with early-stage PDAC and 24 healthy controls. They identified 14,716 differentially methylated regions (DMRs) in plasma cfDNA from PDAC patients compared to controls. They also investigated DNA methylation profiles in microdissected PDAC cells and adjacent normal cells, and found that signals in overlapping plasma cfDNA and tissue DNA methylation were correlated, indicating that cfMeDIP-seq of cfDNA could detect tumor-derived DNA events in ctDNA. The authors then selected the top 300 DMRs (150 hypermethylated and 150 hypomethylated) and assessed their performance compared to a 199-sample validation cohort (35 acute myeloid leukemia, 55 lung cancer, 47 PDAC and 62 healthy controls). They revealed that this approach could discriminate PDAC from other samples with a high value of AUC (0.918). Additionally, the performance was similar between early- and late-stage samples, suggesting applicability to the detection of early-stage PDAC.

The investigation of differential hydroxymethylation in cfDNA was attempted in order to detect early-stage PDAC by determining 5-hydroxymethylcytosine (5hmC) changes. Guller et al. compared 5nhC densities in cfDNA from patients with PDAC to non-cancer controls [38]. They found 5700 hyper- and 6155 hypo-hydroxymethylated genes in 41 PDAC samples compared to 38 non-cancer samples (discovery data set). These altered genes were significantly associated with pancreatic development or functions and cancer pathogenesis. They performed regularized logistic regression analysis, employing top-65% genes with the most variable 5hmC density in cfDNA samples from the discovery data set, which yielded a high value of AUC (0.919). They next tested this regularized regression model in an independent two-fold validation data set—first data set: 23 PDAC and 205 non-cancer samples; and second data set: 7 PDAC and 10 non-cancer samples—and showed a classification performance AUC of 0.921 and 0.943, respectively. Since these validation data sets included more than 50% early-stage (stage I and II) PDAC, the authors suggested that 5hmC changes enable classification of PDAC even during the early stage. 

### 3.3. Circulating microRNA

MicroRNA (miRNA) is a small non-coding RNA composed of 18-25 nucleotides which regulates hundreds of mRNA by inhibiting the translation of specific genes through binding to the 3’-untranslated regions (UTRs) of their target [39]. It is shown to play an important role in tumorigenesis and/or development of various carcinomas including PDAC [39,40]. In addition, miRNAs are stably detectable in the plasma/serum because they are protected from RNase activity by microvesicles such as exosomes [41,42], forming protein complexes with Ago2 [43], and lipoprotein complexes [44]. The stability of miRNAs in body fluids indicates that circulating miRNAs could be useful diagnostic markers in PDAC. In PDAC patients, numerous miRNAs such as miR-21, miR-155, miR-196a, miR-210 and miR-483-3p have been detected in the plasma/serum, and have been suggested as useful biomarkers for diagnosis [45,46,47,48,49]. However, the diagnostic ability of these miRNAs, especially for early-stage PDAC, was not clarified. Ganepola et al. compared differential expressions of plasma miRNAs between early-stage (stage IIA and IIB) PDAC patients (*n* = 8) and healthy controls (*n* = 11) using microarray analysis [50]. They identified miR-642b, miR-885-5b and miR-22 as highly expressed miRNAs in plasma from PDAC patients compared to healthy controls by both microarray and quantitative PCR (qPCR). These three miRNAs were then validated and evaluated as a diagnostic panel with a new cohort of patients (Stage IIA and IIB, *n* = 11) and controls (*n* = 11) and showed high sensitivity (91%) and specificity (91%) for PDAC diagnosis. However, the diagnostic ability for early detection of PDAC by evaluating these miRNAs in plasma is unclear, since the authors did not analyze these three miRNAs in other stages of PDAC than stage II, although these miRNAs could be detected in plasma samples from early-stage PDAC patients [50]. Li et al. measured 735 miRNAs in serum from patients with PDAC (3 cases of stage I and 16 cases of stage II) and controls by miRNA arrays and selected 18 miRNA candidates for validation [51]. Serum levels of 18 miRNAs were examined with individual TaqMan assays for each miRNA in serum from 41 cases with resectable PDAC (stage I: 6; stage II: 28; stage III: 7) and 72 controls (chronic pancreatitis (*n* = 35), pancreatic neuroendocrine tumor (*n* = 18) and 19 healthy controls). Among 18 miRNAs, miR-1290 was the best miRNA that discriminated PDAC from controls. They also compared the diagnostic performance of miR-1290 to CA19-9 and found that miR-1290 had higher overall diagnostic accuracy. Intriguingly, combining these two markers did not improve diagnostic accuracy [51]. Tavano et al. evaluated the plasma level of miR-1290 in 167 PDAC (30 cases with stage I and II, 56 cases with stage III and 73 cases with stage IV) and 267 healthy subjects using ddPCR [52]. They revealed that plasma levels of miR-1290 were higher than controls, but these miRNA concentrations could not differentiate between PDAC patients with single or multiple risk factors for developing PDAC. In addition, CA 19-9 showed better diagnostic performance than miR-1290 and the combination of two markers improved the diagnostic ability for PDAC [52]. Similarly, Wei et al. demonstrated that serum miR-1290 levels were elevated in patients with PDAC (*n* = 120) compared to controls (40 benign disease controls and 40 healthy controls) [53]. Among 20 cases with stage I-II, 6 cases (30%) showed elevated levels of miR-1290, while 54 cases (54%) with stage III-IV expressed high levels of this miRNA [54]. From these results, plasma miR-1290 is a candidate marker for early detection of PDAC, although further studies are required. 

### 3.4. Salivary Molecule Analyses

Saliva is a suitable substance for disease screening because it is easily and noninvasively obtained from patients. Thus, germline genetic alterations in samples from saliva were employed for hereditary cancer risk assessment [54]. In addition, salivary mRNA is reported to be relatively stable and informative for disease diagnosis, including cancer [55,56]. For diagnosis of PDAC, molecular investigations using saliva have been performed. To explore whether analysis of salivary mRNA could lead to diagnosis of early-stage PDAC, differential expression of salivary mRNA was examined in 114 samples (12 resectable PDAC and 12 healthy control samples for discovery phase; 30 patients with resectable PDAC, 30 chronic pancreatitis patients and 30 healthy control subjects for the biomarker validation phase) [57]. Candidate biomarkers identified from microarray were first verified using the discovery sample set. Then, 49 up-regulated and 21 down-regulated transcripts in PDAC samples relative to controls were selected by microarray and validated by qPCR. Among these RNAs, qPCR confirmed that 23 up-regulated and 12 down-regulated transcripts were consistent with the microarray data. These candidates were then subjected to validation by qPCR in an independent cohort of 30 patients with PDAC, 30 healthy controls and 30 chronic pancreatitis patients. A total of 7 up-regulated (*MBD3L2, KRAS, STIM2, DMXL2, ACRV1, DMD* and *CABLES1*) and 5 down-regulated (*TK2, GLTSCR2, CDKL3, TPT1* and *DPM1*) genes were validated by the results of qPCR. These 12 mRNA all showed significant differences in the expression level between PDAC and healthy controls or chronic pancreatitis. The logistic regression model with a combination of 4 mRNA (*MBD3L2, KRAS, ACRV1* and *DPM1*) was able to discriminate patients with PDAC from noncancer subjects, yielding a high value of AUC (0.971) with 90.0% sensitivity and 95.0% specificity. Xie et al. investigated whether salivary miRNAs could be a suitable biological marker for detection of resectable PDAC patients [58]. Saliva samples were obtained from 48 patients with resectable PDAC, 20 patients with benign pancreatic tumors and 48 healthy controls. Among these 116 samples, 8 PDAC and 8 healthy control samples were selected for the discovery phase. They selected 10 candidate miRNAs using miRNA microarray analysis in the discovery set. They next examined the expression level of the 10 miRNAs by qPCR in the discovery set as a verification phase. In this phase, miR-3679-5p was significantly down-regulated and miR-940 was significantly up-regulated in PDAC patients compared to controls, while the other 8 miRNAs did not show any significant difference. In the validation phase, the two verified miRNAs were analyzed using qPCR with the independent sample set of 40 PDAC, 20 benign pancreatic tumors and 40 healthy controls. The results showed that miR-3679-5p and miR-940 were significantly down- and up-regulated in PDAC compared to noncancer (healthy controls and benign pancreatic cancer), respectively. The combination of the two validated miRNAs were used to construct binary logistic regression models and this model yielded an AUC of 0.763 with 70.0% sensitivity and 70.0% specificity (PDAC vs. noncancer). The same group evaluated the expression of salivary long non-coding RNA (lincRNA) in patients with PDAC (*n* = 55: 9 stage I, 14 stage II and 32 stage III) and compared to healthy controls (*n* = 55) and benign pancreatic tumors (*n* = 20) [59]. They identified HOTAIR and PV1T as significantly up-regulated lincRNA in PDAC group compared to controls and benign pancreatic tumors. The combination of salivary HOTAIR and PVT1 differentiated PDAC from healthy controls with sensitivity of 78.2% and specificity of 90.9%. This aggregation set differentiated PDAC from benign pancreatic tumors, with sensitivity of 81.8% and specificity of 95.0%. 

## 4. Conclusions

The candidate biomarkers in body fluids for early detection of PDAC are summarized in Table 1. The most effective strategy for pancreatic screening to improve the prognosis of PDAC patients is to identify the lesion prior to invasive PDAC. However, the detection of high-grade PanIN, which is a precursor to invasive cancer, is almost impossible by imaging methods. To date, accumulated evidence suggests that molecular analyses using body fluids could be useful for diagnosing PDAC. Therefore, many molecular approaches by liquid biopsy have been attempted to determine whether they could detect preinvasive lesions in individuals. Measurement of *TP53/SMAD4* concentrations in pancreatic juice by digital NGS is likely to be the most feasible marker to detect high PanIN lesions, while the collection of pancreatic juice is relatively invasive compared to obtaining blood or saliva. The relative invasiveness and difficulty of collecting pancreatic juice might be a limitation of this assessment. Combination analysis of various gene mutations and protein markers in blood samples, as performed in CancerSEEK, would be a candidate screening method for early detection of PDAC but is unlikely to be a candidate method for precursor lesion. Since the number of circulating tumor cells is very small before a tumor becomes invasive, it is likely to be challenging to discover preinvasive lesions by molecular evaluation in plasma and/or saliva. Although high diagnostic values for PDAC, including early stages, were observed in the analyses of epigenetic alteration and miRNA expression in blood samples, further large-scale study would be required to confirm whether these approaches could have clinical utility, because the number of early stages in this study was small.

## Figures and Tables

**Table 1 diagnostics-11-00375-t001:** Summary of candidate biomarkers for early detection of pancreatic ductal adenocarcinoma in body fluids by molecular approaches.

**Materials**	**Candidate Biomarker**	**Performance for Early Detection of Pancreatic Ductal Adenocarcinoma (PDAC)**
**Pancreatic juice**	*KRAS*	Detection of PDAC with approximately 60–70% sensitivity and 80% specificity [11,18]. Limitations: difficulty in differentiating early-stage PDAC from IPMN and/or chronic pancreatitis.
	*TP/53/SMAD4* concentration	Discrimination of PDAC or high-grade dysplasia from all other subjects with 61.1% sensitivity and 95.7% specificity (AUC = 0.819) [20,21]
**Blood**	*KRAS*	Detection of early-stage PDAC with 30 to 70% frequency. Limitation for early detection; lower ability to detect early stage compared to advanced stage; difficulty in avoiding false positive [25,26,27,28,29,30,31,32].
	Combination assay of gene mutation and protein biomarkers; The presence of mutation in 16 genes *(NRAS, CTNNB1, PIK3CA, FBXW7, APC, EGFR, BRAF, CDKN2A, PTEN, FGFR2, HRAS, KRAS, AKT1, TP53, PPP2R1A* and *GNAS)* or an elevation in the level of any of 8 proteins (CA125, CEA, CA19-9, Prolactin, HGF, OPN, Myeloperoxidase and TIMP-1)	Detection of early stage (stage I-III) with more than 70% sensitivity and 99% sensitivity [29]
	Methylation of *BNC1* and *ADAMTS1*	Detection of 100% (8/8) of stage I, 88.9% (8/9) of stage IIA and 100% (20/20) of stage IIB PDAC patients, with combined sensitivity of 97.3% and specificity of 91.6%, by combination of the two genes [36]
	miR-1290	Detection of stage I-II PDAC with 30% frequency [52]
**Saliva**	Combination of HOTAIR and PVT1	Differentiation of early PDAC (stage I-III, *n* = 55) from healthy controls (*n* = 55), with sensitivity of 78.2% and specificity of 90.9% [57].

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
