# Peer review of "Molecular Approaches Using Body Fluid for the Early Detection of Pancreatic Cancer"

_diagnostics, 2021, doi:10.3390/diagnostics11020375_

Round 1
Reviewer 1 Report
The review is informative, but poorly structured and many typos.
Line 48 - maybe this is point 2.
In general, I would rather name the sub-points on the biological fluid under study. Can genetic mutations be detected in saliva?
Lines 357-421 - Template not deleted.
In conclusion, it is necessary to discuss the limitations of the use of these biological fluids in more detail.
Author Response
I very much appreciate for your comments and suggestions.
I have re-edited the manuscript according to your comments and suggestions and responses to your comments and suggestions are summarized as below.
# Line 48 - maybe this is point 2
I very much appreciate for your identification. I have corrected “1.2” to “2 “in line 48.
# In general, I would rather name the sub-points on the biological fluid under study.
I very much appreciate for your suggestion. I have corrected “4” to “3.4” in line 284.
# Can genetic mutations be detected in saliva?
Germline mutations have found in samples from saliva and have been examined for hereditary cancer risk assessment including pancreatic cancer. Thus, I have added this description (lines 288 to 289) and reference (ref no. 54).
# Lines 357-421 - Template not deleted.
I very much appreciate for your identification. I have deleted template.
# In conclusion, it is necessary to discuss the limitations of the use of these biological fluids in more detail.
I very much appreciate for your suggestion. I have added the limitations and related description in the conclusion (Lines 344-345, and, lines 347 to 349).

Reviewer 2 Report
This paper is a very well-organized review for the early detection of pancreatic cancer. Here are some minor comments.
- Line 48, 1.2. Gene mutation analysis in pancreatic juice -> 2. Gene mutation analysis in pancreatic juice
- Line 357 2. Material and Methods -> remove
- Table 1. Please match the rows and columns. It is difficult to see.
- Table 1. Candidate biomarkers: KRAS is duplicated.
- Serum biomarker signature is a tenable approach to detecting early state PDAC (J Clin Oncol 36:2887-2894). Could you introduce the related studies?
Author Response
I very much appreciate for your comments and suggestions.
I have re-edited the manuscript according to your comments and suggestions, and responses to your comments and suggestions are summarized as below.
- Line 48, 1.2. Gene mutation analysis in pancreatic juice -> 2. Gene mutation analysis in pancreatic juice
I very much appreciate for your identification. I have made correction in line 48.
- Line 357 2. Material and Methods -> remove
I very much appreciate for your identification. I have removed the section you pointed out.
- Table 1. Please match the rows and columns. It is difficult to see.
I very much appreciate for your comments. I have re-edited the Table 1 in the revised manuscript.
- Table 1. Candidate biomarkers: KRAS is duplicated.
I very much appreciate for your identification. I would like to mean that KRAS is candidate biomarker in both pancreatic juice and blood. Thus, I have description the materials as bold to avoid confusion.
- Serum biomarker signature is a tenable approach to detecting early state PDAC (J Clin Oncol 36:2887-2894). Could you introduce the related studies?
I have introduced this excellent study in line 151 to 153 and added the reference (ref no. 30).

Round 2
Reviewer 1 Report
The authors responded to all the comments of the reviewer and made corrections to the text of the manuscript. I believe that the article in its current form can be accepted for publication.